# Textiles for Circular Fashion: The Logic behind Recycling Options

**Paulien Harmsen** * , **Michiel Scheffer and Harriette Bos**

Wageningen Food and Biobased Research, P.O. Box 17, 6700 AA Wageningen, The Netherlands;
michiel.scheffer@wur.nl (M.S.); harriette.bos@wur.nl (H.B.)
* Correspondence: paulien.harmsen@wur.nl

**Abstract:** For the textile industry to become sustainable, knowledge of the origin and production of resources is an important theme. It is expected that recycled feedstock will form a significant part of future resources to be used. Textile recycling (especially post-consumer waste) is still in its infancy and will be a major challenge in the coming years. Three fundamental problems hamper a better understanding of the developments on textile recycling: the current classification of textile fibres (natural or manufactured) does not support textile recycling, there is no standard definition of textile recycling technologies, and there is a lack of clear communication about the technological progress (by industry and brands) and benefits of textile recycling from a consumer perspective. This may hamper the much-needed further development of textile recycling. This paper presents a new fibre classification based on chemical groups and bonds that form the backbone of the polymers of which the fibres are made and that impart characteristic properties to the fibres. In addition, a new classification of textile recycling was designed based on the polymer structure of the fibres. These methods make it possible to unravel the logic and preferred recycling routes for different fibres, thereby facilitating communication on recycling. We concluded that there are good recycling options for mono-material streams within the cellulose, polyamide and polyester groups. For blended textiles, the perspective is promising for fibre blends within a single polymer group, while combinations of different polymers may pose problems in recycling.

**Keywords:** textile; recycling; circular fashion; polymer structure

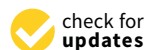



## 1. Introduction

The current textile industry uses large amounts of non-renewable resources and applies hazardous substances and polluting processes. On top of this, the ever-spreading trend of fast fashion has led to fast-fashion retailers selling clothing expected to be discarded after being worn only a few times [1]. Hence, new solutions to reduce the use of (virgin) resources need to be developed and implemented. One of the proposed routes is to increase circularity in the textiles and clothing industry [2].

However, what is the definition of circularity in relation to the textile industry? Circularity is a concept that originates from the field of industrial ecology, combined with circular design concepts such as cradle to cradle [3,4]. The underlying concerns are the ever-increasing depletion of non-renewable feedstock. Circular solutions thus aim at fulfilling societal demand while minimising the input of virgin resources. Circular concepts, therefore, go much further than just recycling materials into new products. As stated by Rosa et al., an actual transition towards a Circular Economy requires relevant changes along the whole value chain, not only for waste generation and resource use but also for adopted market strategies and business models [5].

Several circularity strategies exist to reduce the consumption of natural resources and materials and minimise waste production. They can be prioritised according to their levels of circularity, i.e., the 10R-strategy [6]. The hierarchy of circular solutions relevant for the

textiles and clothing market segments can be presented in Figure 1, which was built on this 10R-strategy.

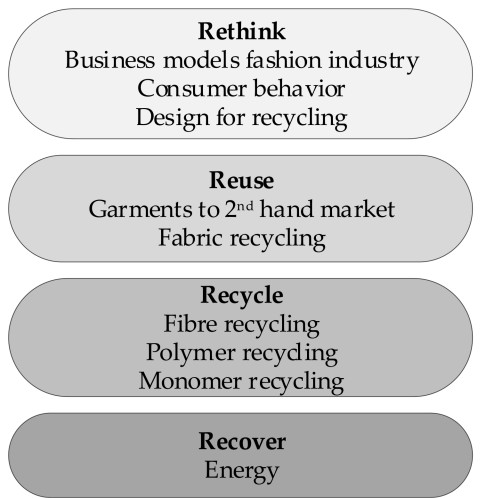

**Figure 1.** Different options for circular textiles based on the 10R-strategy. The higher in order, the more preferable and sustainable the options are.

Ideally, waste is prevented by changing consumer buying behaviour, wearing clothes for a longer period, implementing other business models in the fashion industry such as textile rentals or by applying design for longevity principles (Rethink). Ideally, when a garment is discarded, it is reused by another customer through the second-hand market. When the garment is no longer wearable, it can be converted into a product of lower value, such as wiping rags (Reuse). When the fabric is no longer usable as such, recycling techniques come into play. Here, a distinction can be made between fibre, polymer or monomer recycling (Recycle). When recycling is also no longer possible, recovery of energy (Recover) is the final option.

In this publication, we focus mainly on the Recycle part. According to Directive 2008/98EC [7], recycling means 'any recovery operation by which waste materials are reprocessed into products, materials or substances, whether for the original or other 'purposes' [8]. The emphasis in our publication is on the options for recycling textile fibres back into textiles, but other end-uses of recycled fibres are also mentioned.

Initiatives and technologies for textile recycling are emerging rapidly, and similarities with plastic recycling are apparent. However, for textiles, the large variety of materials used (i.e., fibres, auxiliaries) combined with a high level of structural complexity is substantially different. For a better understanding of developments in textile recycling, we encountered three fundamental problems.

The first is the definition of the classes of textile fibres. The various fibre types are classed as either being natural (vegetable, animal, mineral) or artificial (regenerated/natural, synthetic and inorganic) [9,10]. However, it is much wiser for recycling purposes to classify the fibres based on their main chemical bonds instead of their origin, as fibres with the same kind of chemical bonds usually have similar chemical and, often, physical characteristics. In this publication, we propose a new classification of textile fibres that matches recycling technologies.

A second issue is the classification of recycling technologies. There is no standard definition, and many descriptions are used for textile recycling [11]. Sandin and Peters propose a "topology of textile reuse and recycling" based on the level of disassembly of the recovered material (fabric, fibre, polymer/oligomer and monomer recycling) [12]. They state that the "systematisation of recycling routes into mechanical, chemical and thermal ones is ambiguous and questionable". We support this statement, but at the same time, we argue that the classification based on disassembly is also not entirely suitable. In this paper,

we combine the two approaches into one new classification of textile recycling based on the level of disassembly combined with mechanical, physical and chemical recycling methods. This approach is based on the molecular structure and structure/properties relation of the polymers that make up the fibres, enabling us to unravel the logical and preferred recycling routes for the various types of textile fibres.

The last problem we face is the lack of clear communication about (1) the technological progress by companies and brands and (2) the benefits of textile recycling from a consumer perspective. Wagner and Heinzel state that "consumers' awareness, attributed importance, and perceived value become crucial for the success of recycled synthetic textile products" [11]. The choice between virgin and recycled textile products depends on the perceived benefits or disadvantages. Recycled content is often thought to be synonymous with low quality, but we show that the final quality strongly depends on the type of fibre and the technology used.

The main aim and novelty of this work is the design of improved classification systems for textile fibres and textile recycling. Our hypothesis is that an approach based on polymeric structures contributes to a better understanding of textile recycling and a better definition of the challenges ahead.

We explain our applied methods in Section 2. In Section 3, our main results are presented, i.e., a new classification system of textile fibres and textile recycling, with the corresponding recycling methods (mechanical, physical and chemical). This section is complemented with a schematic overview of resources and processes to produce garments in a Linear Fashion manner based on virgin resources or in a Circular Fashion manner with recycling and reuse. Section 4 is dedicated to discussions on recycling options for the main polymer types defined, both from a theoretical approach and based on commercial activities published on websites. Based on this discussion, a feasibility assessment was created, followed by a discussion on the challenge of blended textiles. Finally, Section 5 provides the limitations of our study, some concluding remarks and suggestions for further actions to be taken.

## 2. Methods

A new classification system for textile fibres that combines well with the various recycling methods available was designed. This classification system was set up by first determining the primary textile fibres from Mather [9], their corresponding polymeric structures and main chemical linkages. This resulted in six polymer types that are reviewed throughout this publication: cellulose, polyester, polyamide, polyurethane, polyolefin and polyacrylic. The following step was placing the textile fibres in their corresponding polymer category. As a final step, the type of polymer (a natural, addition or condensation polymer) was also included, which is relevant for recycling.

In addition, a new classification system for textile recycling was designed. We adopted a systematisation of recycling routes presented by the Ellen MacArthur Foundation [2] and Sandin and Peters [12], which is based on the level of disassembly: recycling to fibre, polymer or monomer. Each of these recycling routes requires (several) subsequent recycling methods to arrive at the desired product. With the polymeric structure of the fibres in mind, we distinguished sorting, mechanical, physical and chemical methods. As a final step, the classification of textile recycling was coupled with the appropriate recycling methods.

When the linear fashion chain transforms into a circular fashion chain, resources will partly change from virgin to recycled content. How this recycled content changes the textile supply chain is illustrated by one schematic. To this end, the key stages in the apparel life cycle were gathered from Tomaney [13], and linked to the output streams from recycling (on the level of fibres, polymers and monomers).

The discussion section consists of three parts. The applicability of the three recycling routes (fibre, polymer, monomer) was determined for the six polymer types. The physical structure/property relations and chemical options to break down the primary polymeric bonds were determined by desk research. The polymeric structure of the fibres and the

main bonds in the polymer backbone served as a guideline to determine which methods are potentially applicable and which are not realistically possible. The potentially available methods were illustrated by the recycling initiatives that resulted from our (scientific and grey) literature review.

Based on these results and our own judgement, we constructed a feasibility assessment for the main textile polymers and fibres, emphasising options for textile to textile. For this assessment, we assumed that all textiles consist of only one type of fibre (mono-material textiles).

However, textiles are often not composed of one type of fibre but of a variety of fibres, i.e., blended textiles. The recycling classification system and feasibility assessment were used to explore the recycling options for these blended textiles. An estimation for common textile blends was presented, including an outline of current recycling initiatives in this field.

## 3. Results

### 3.1. Classification of Textile Fibres

Textiles mainly consist of fibres, i.e., very long (2–5 cm) and very thin structures. Landi [14] describes fibre as a "unit matter with a length at least 100 times its diameter, a structure of long-chain molecules having a fixed preferred orientation, a diameter of 10–200 microns, and flexibility". Thus, all fibres have a molecular structure that contributes to their specific attributes and properties. All fibres are characterised by the common characteristics of small diameter relative to its length, flexibility and fineness [10].

Classification of textile fibres can be done in many ways. A widely used method to classify fibres is based on their origin and resources, i.e., animal, vegetable, regenerated or synthetic (mainly fossil-based) fibres. Although the origin of textile fibres is an important parameter, we propose a new classification for the definition of recycling options and the circular use of textiles. This classification is based on chemical groups and bonds that form the backbone of the polymers that the fibres are made of and that impart characteristic properties to the fibres. For recycling purposes, it is useful to classify the various types of textile fibres based on their main chemical bonds, as fibres with the same chemical bonds usually have similar chemical and often physical characteristics. Regarding textile recycling, this classification method is preferred, as it helps in evaluating the options that are available in preventing the generation of textile waste. Table 1 shows the main polymer groups discussed in this publication, i.e., polysaccharides (i.e., cellulose), polyesters, polyamides, polyurethanes, polyolefins and polyacrylics, and presents the chemical links in the polymeric backbone that are relevant for recycling approaches.

**Table 1.** Classification of textile fibres based on polymer linkages.

| Polymer | Polysaccharides: Cellulose | Polyester | Polyamide | Polyurethane | Polyolefin | Polyacrylic |
|---|---|---|---|---|---|---|
| Essential linkage [1] | β-glycosidic  | ester  | amide  | urethane  | alkane  | acrylonitrile  |
| Fibre examples | Cotton (natural) Linen (natural) Viscose (natural) Lyocell (natural) | PET [2] (condensation) | Wool (natural) Silk (natural) Nylon (condensation) | Elastane (condensation) | PP [2] (addition) PE [2] (addition) | Acryl (addition) Modacryl (addition) |
| Melting point | No | Yes | No (natural) Yes (condensation) | Yes | Yes | No |

[1] Only for cellulose is this the repeating unit; for the other polymers, it denotes the linkage relevant for recycling. [2] PET: polyethylene terephthalate, PP: polypropylene, PE: polyethylene.

Polymers can also be categorised as being natural, addition or condensation polymers. For the main textile fibres, this is also included in Table 1. Natural polymers are of biological origin and are formed by living organisms such as plants or animals. Cellulose (polysaccharide) and protein (polyamide) are examples of natural polymers relevant for textiles. Addition and condensation polymers are synthetic polymers built up of monomeric building blocks. Condensation polymers are formed by the reaction of two different functional groups, usually originating from two or more different monomers. During the polymerisation reaction, small molecules such as water are often eliminated. Examples include polyesters, polyamides and polyurethanes. Addition polymers are formed mainly by one type of monomer. The polymerisation reactions are chain reactions, with free radicals or ionic groups responsible for propagating the chain reaction. Examples include polyolefins and polyacrylics [15].

### 3.2. Categorising Textile Recycling and Recycling Methods

Textile recycling includes all processes on the level of fibres, polymers or monomers. Whether a garment is suitable for fibre, polymer or monomer recycling is determined in large part by the fibre composition and the chemical structure of the polymers that make up the fibres.

- Fibre recycling implies the preservation of the fibres after the disintegration of the fabric.
- Polymer recycling includes the disassembly of the fibres while the polymers remain intact.
- Monomer recycling implies that fibres and polymers are broken down into their chemical building blocks.

Each of these recycling methods requires (several) subsequent recycling methods to come to the desired product. We distinguish sorting, mechanical, physical and chemical methods. Because stakeholders do not always interpret terms such as mechanical and chemical in the same way, we propose the following definitions that help us present the different recycling methods for textiles in an orderly and recognisable fashion:

- Mechanical methods break down the fabric and retain the fibres by cutting, tearing, shredding or carding. The fibre length is reduced as an unwanted side effect, thereby affecting the spinnability and yarn strength [16]. The fibres will have a shorter fibre length than the original fibres, and some dust will be generated [17].
- Physical methods use physical processes to make the fibres or polymers suitable for reprocessing, either by melting or dissolving them. With physical recycling, the structure of the fibres is changed, but the polymer molecules that make up the fibres remain intact. After melting or dissolving, either melt spinning or solution spinning can be used to form a new filament (i.e., a fibre of infinite length).
- Chemical methods exploit chemical processes to break down fibres and polymers. The polymers that make up the fibres are either modified or broken down, sometimes to their original monomeric building blocks. This can be done by chemical or biological methods (e.g., with enzymes). After chemical recycling, the building blocks can be repolymerised into a new polymer.

In Table 2, the classification of textile recycling is coupled to the corresponding methods. It is evident that for all recycling initiatives, extensive sorting is needed, as mixtures of different materials and colours will result in recycled feedstock of poor quality. Mechanical methods are often required; for fibre recycling, this is the only step, but for polymer and monomer recycling, it is a treatment prior to physical or chemical recycling methods.

**Table 2.** Classification of textile recycling and corresponding recycling methods.

| Classification of Textile Recycling | Recycling Methods | |
| --- | --- | --- |
| Fibre recycling | Mechanical methods | |
| Polymer recycling | Mechanical methods | Physical methods |
| Monomer recycling | Mechanical methods | Chemical methods |

### 3.3. Resources, Production Methods and Recycling Routes in the Linear and Circular Textile Industry

A schematic overview of the relationship between resources, intermediate products and various processes applied in the textile industry is presented in Figure 2. Several subsequent standard processing steps are needed to make a garment from renewable or fossil resources.

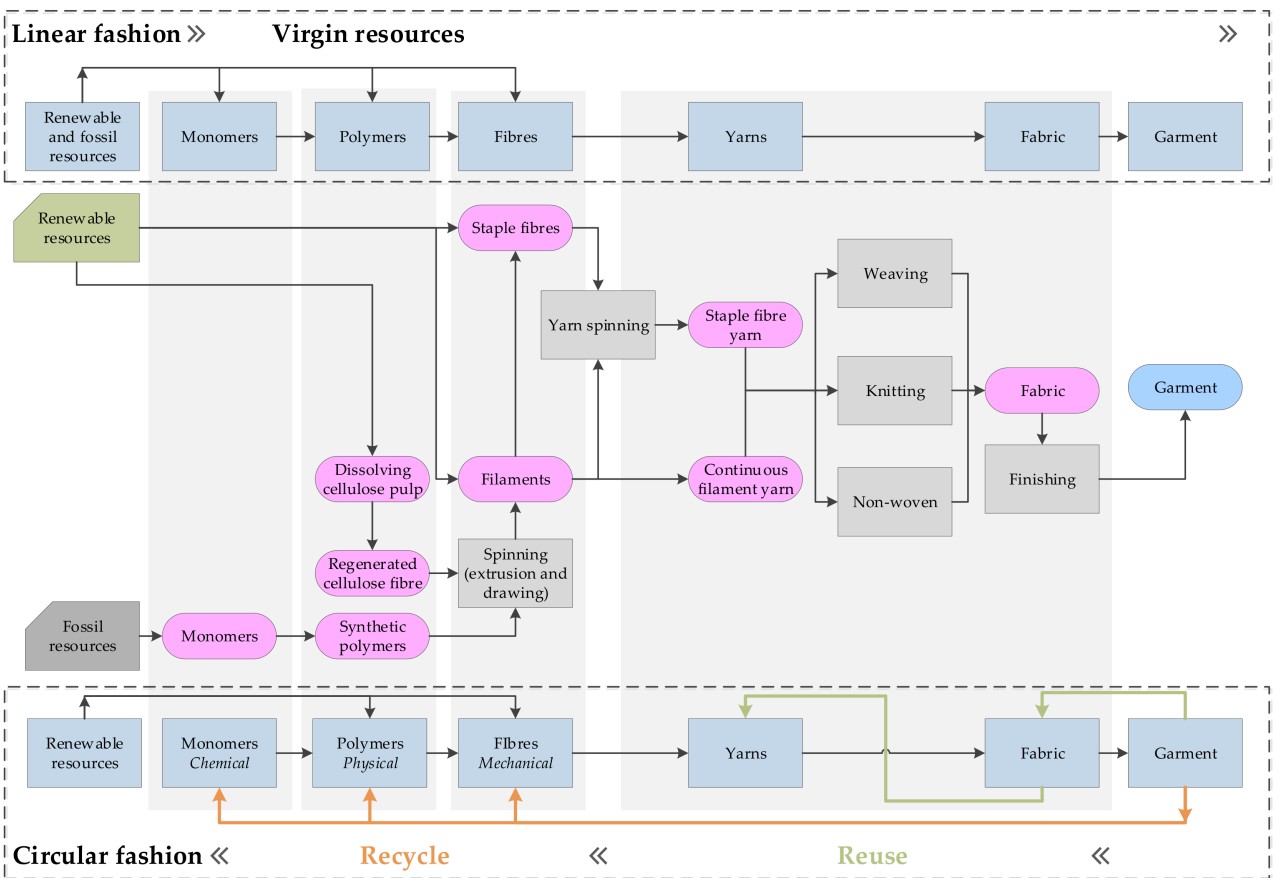

**Figure 2.** Relation between resources and processes to produce garments. Above is the linear route, below is the circular route.

In the current linear fashion chain, virgin resources can be either of renewable origin, coming from plants or animals, or fossil-based (Figure 2, upper and middle part). Fossil resources always enter the scheme on the monomer level, as the resulting synthetic fibres are built-up of monomeric building blocks that are polymerised into polymers. Synthetic polymers are spun into a fibre shape by extrusion spinning, either from the melt or the solution, and then spun into a yarn. Renewable resources enter the scheme on the polymer or fibre level, not yet on the monomer level. In a future scenario, this may change, as it is also possible to make synthetic polymers from renewable resources. On the polymer level, we find the natural polymer cellulose, mainly harvested from wood to produce cellulose pulp. This cellulose is subsequently spun into a fibrous shape by extrusion spinning, after which regenerated cellulose fibres are formed (continuous filaments), e.g., viscose or lyocell. On the fibre level, we find natural fibres that can be staple fibres (cotton, wool, linen) or filaments (silk), and they can be yarn spun after minor processing.

In the circular fashion chain, resources come from renewable resources and garments (post-consumer) or residues produced in the factories in the form of yarns or fabrics (post-industrial) [18] (Figure 2, lower part). These post-industrial and post-consumer residues are not yet optimised for recycling. There are distinct differences between these streams. It is widely recognised that the post-consumer stream forms the most challenging one

(not well-defined, contaminated, decreased fibre quality due to washing and wearing). In addition to the recycled input, virgin renewable resources will always be required to overcome quality loss when polymer and fibre recycle streams are applied, in contrast to the recycling route via monomers, as (re)starting from the monomer level will always result in virgin quality fibres.

As shown in Figure 2, monomer, polymer and fibre recycling each follow a specific pathway and re-enter the textile production cycle at a different level. The preferred entrance level is highly dependent on the type of material that needs to be recycled. The overall aim is to keep the structure of the materials intact as much as possible and to minimise processing.

## 4. Discussion

*4.1. Recycling Methods for Six Polymer Types: Theoretical Approach and Commercial Activities*

This section presents the applicability of the three different recycling approaches for the six polymer types: cellulose, polyester, polyamide, polyurethane, polyolefins and polyacrylics.

### 4.1.1. Cellulose

Only two routes are available for cellulose textiles when new textile fibres are envisioned as the end-product: mechanical recycling or physical recycling. A third option, the breakdown of cellulose to glucose molecules by chemical or enzymatic methods, is currently less relevant for textiles.

- Mechanical recycling to fibres Table 3 shows that several mechanical cellulose recycling factories produce a variety of products and use cotton as input. Mechanical recycling of cotton yields fibres typically applied in the nonwoven industry and as flock (very small fibres used to create texture on surfaces). Recycled cotton fibres are shorter than virgin ones and thus more difficult to spin. Mechanically recycled fibres are often mixed with (longer) virgin fibres such as PET or cotton for woven applications. About 95% of the recovered fibres by the mechanical recycling of cotton are directly processed into nonwovens for the automotive industry, appliances, drainage systems and geotextiles [17].
- Physical recycling to polymers Polymer recycling of cellulose materials to yield regenerated cellulose fibres is an excellent option to recover and reuse cellulose polymers from residue streams. The resource to produce regenerated cellulose is usually wood, but other cellulose-containing resources could also be used, for example, postconsumer textiles. Possible bottlenecks are the contamination of the garments with other types of fibres and the presence of finishing agents and dyes. Cotton fibres are almost entirely made up of cellulose. They are excellent candidates as feedstock for the viscose and lyocell process, and this is applied on a small scale by several parties.

**Table 3.** Cellulose recycling initiatives (partly adapted from [19]).

| Company | Input Stream | Product | Status | Ref. |
|---|---|---|---|---|
| **Mechanical recycling to fibres** | | | | |
| Frankenhuis | Post-consumer textile waste (cotton) | Textile fibres for nonwovens | Commercial, the Netherlands | [20] |
| Wolkat | Post-consumer textile waste (cotton) | Yarns for textiles and nonwovens | Commercial, the Netherlands and Morocco | [21] |
| Belda Llorens | Cotton waste a.o. | EcoLife yarns, blended with virgin, for textiles | Commercial, Spain | [22] |
| Geetanjali Woollens | Cotton waste a.o. | Recycled cotton fibre and yarn, blended with virgin for textiles | Commercial, India | [23] |
| Ferre | Cotton waste | Recover yarns, blended with virgin for textiles | Commercial, Spain | [24] |
| Velener Textil GmbH | Post-industrial cotton yarns | WECYCLED® cotton fibres, blended with virgin for textiles | Commercial, Germany | [25] |
| **Physical recycling to polymers** | | | | |
| Lenzing AG with Refibra™ (lyocell) | 20% post-industrial cotton scraps combined with 80% virgin cellulose pulp from wood | Tencel™ lyocell fibres (regenerated cellulose fibres) | Commercial, Austria | [26] |
| Asahi Kasei | 100% cotton linter, post-industrial residue of cotton processing | Bemberg™ cupro fibre (regenerated cellulose fibres) | Commercial, Japan, 17.000 mt/y | [27] |
| Renewcell | High-content cellulose waste (cotton, regenerated cellulose) | Dissolving pulp (Circulose pulp) | Demonstration, Sweden, 7000 mt/y | [28] |
| Evrnu with NuCycle™ | High-content cellulose waste (cotton, regenerated cellulose) | Regenerated cellulose fibres | In development, USA | [29] |
| Infinited Fibre | High-content cellulose waste (cotton, regenerated cellulose) | Regenerated cellulose fibres | In development, Finland | [30] |
| Aalto University with Ioncell™ | High-content cellulose waste (cotton, regenerated cellulose) | Ioncell™ fibres (regenerated cellulose fibres) | In development, Finland | [31] |
| SaXcell with SaXcell® (lyocell) | High-content cellulose waste (cotton, regenerated cellulose) | SaXcell® fibres (regenerated cellulose fibres) | In development, the Netherlands | [32] |

### 4.1.2. Polyester

The term 'polyester' is a generic name for all polymers containing ester linkages in their polymeric chain, but in the apparel sector, 'polyester' stands for one of these types of materials: polyethylene terephthalate (PET). The recycling of PET has become increasingly important with the increasing use of PET in textiles. For textiles, PET is most suited for physical and chemical recycling, and mechanical recycling is less achievable.

- Physical recycling to polymers PET is a thermoplastic material; it melts at elevated temperature (>260 °C) and can be re-spun into fibres again. Contaminations may pose a problem in these kinds of processes, and this is therefore the reason that recycled fibres for textiles are often produced from transparent bottles (Table 4). Recycling bottles into fibres is implemented across the globe. Post-consumer transparent bottles are relatively clean and result in a high-quality rPET (recycled PET), suitable for yarn production. The use of less pure post-consumer PET items such as coloured bottles, trays and films and PET recovered from the ocean and textiles [8,33] is more challenging. If these sources cannot be sufficiently cleaned, it may be necessary to use chemical recycling methods.

- Chemical recycling to monomers and oligomers Chemical recycling methods are well suited for the production of PET fibres (Table 4). During chemical recycling, the polyester molecules are broken down into smaller fragments. These smaller fragments are recovered by separation processes such as filtration, precipitation, centrifugation

and crystallisation [34], making the removal of contaminants easier than in mechanical and physical recycling.

The technology of PET degradation or depolymerisation is done by solvolysis, i.e., reaction in a solvent in which one of the reactants is the solvent molecule. Solvolysis can be divided into various techniques, common for the depolymerisation of polycondensation polymers such as polyesters, polyurethanes and polyamides [35].

**Table 4.** Polyester recycling initiatives (partly adapted from [19]).

| Company | Input Stream | Product | Status | Ref. |
|---------|-------------|---------|--------|------|
| **Physical recycling to polymers** | | | | |
| Velener Textil GmbH | PET bottles | WETURNED® PET-woven fabric | Commercial, Germany | [36] |
| Cumapol with CuRe Technology | Coloured PET from various sources | Transparent PET granulate | In development, the Netherlands | [37] |
| **Chemical recycling to monomers and oligomers** | | | | |
| Ioniqa with glycolysis | (Coloured) PET from various packaging materials | BHET and rPET | Commercial, the Netherlands, 10 ktons/y | [38] |
| Jeplan with glycolysis | (Coloured) PET from various packaging materials | BHET | Commercial, Japan | [39] |
| Teijin | Bottles and other PET materials | DMT to ECOPET™ filament yarns | Commercial, Japan | [40] |
| Eastman with chemical recycling (PRT) | Polyester | Unknown | Commercial, USA | [41] |
| Ambercycle with enzymatic hydrolysis | Post-consumer textile waste | Cycora™yarn for new textiles | In development, USA | [42] |
| Carbios with enzymatic hydrolysis | PET | Monomers EG and TPA | In development, France | [43] |
| Gr3n with microwave radiation | PET | Monomers | In development, Switzerland | [44] |

The production of building blocks by chemical PET recycling can result in the original monomers: ethylene glycol (EG) and terephthalic acid (TPA) [35]. Other processes result in oligomers, i.e., dimers or trimers of the original building blocks, such as bis(2-hydroxyethyl), terephthalate (BHET) or dimethyl terephthalate (DMT) [34].

The most important advantage is that virgin quality PET can be achieved; the disadvantage is that chemical recycling is more expensive and requires large scale production to become economically feasible [34]. Various companies are upscaling and validating their technologies.

### 4.1.3. Polyamide

In contrast to the other polymer categories, the two main polyamide fibre groups, wool (natural fibre) and nylon (synthetic fibre), have very different recycling options, even though they both have amide bonds in their polymer backbone. Wool can be recycled me-

chanically [45], whereas for nylon, being a polycondensation polymer and a thermoplastic material, similar recycling options as for polyesters are viable [35] (Table 5).

- Wool: Mechanical recycling to fibres Recycled wool has a long tradition and is the single example of successful mechanical recycling of post-consumer textiles to new yarns for textiles. The mechanical recycling of wool is the only option for the time being. Woolen fabrics are predominantly made from long fibres and handled with care, making mechanical recycling into fibres achievable. The mechanical recycling of wool is done by similar steps as mechanical recycling of cotton. Sorted fabrics are cleaned and turned into fibres by cutting and tearing. Post-industrial wool processing residues (fibre, yarn and fabric generated during production) are routinely recycled back into the manufacturing process flow for reasons of economic efficiency [45]. Mechanical recycling of post-consumer streams is also feasible. The applicability of post-consumer residues is partly dependent upon effective ways to minimise fibre breakage and maximise residual fibre length after the mechanical pulling process.
- Nylon: Physical recycling to polymers Nylon can be melted and reshaped into new fibres of appropriate length and strength. Nylon waste is available as carpets and fishing nets (composed of nylon 6), and these materials are an important feedstock for physical nylon recycling. Recycling fishing nets is technologically feasible, and recycled nylon 6 shows similar characteristics compared to commercial nylons [46]. Recycling carpets is also possible by blending various carpet components through reactive extrusion and compatibilisation, yielding lower-quality products [47].
- Nylon: Chemical recycling to monomers For nylons, chemical recycling methods are well suited. Nylon 6 is depolymerised to retrieve its original building block caprolactam. The process design is suitable for processing 'contaminated' materials, as the product exits at the top of the reactor and residues remain on the bottom [47]. According to DSM [48], this process is not more expensive than the production of virgin caprolactam, and it is much more environmentally benign.

**Table 5.** Polyamide recycling initiatives (partly adapted from [19]).

| Company | Input Stream | Product | Status | Ref. |
|---------|-------------|---------|--------|------|
| **Wool: mechanical recycling to fibres** | | | | |
| Cardato | Post-industrial Post-consumer | Yarns, "Cardato Recycled" Brand | Commercial, 22.000 mt/y, Italy (Prato) | [19] |
| Boer Group | Post-consumer | Yarns | Commercial, Netherlands | [49] |
| Geetanjali Woollens | Unknown | Recycled sheep wool and cashmere | Commercial, India | [23] |
| Novetex (Billie System) | Various | Unknown | Commercial, Hong Kong | [50] |
| **Nylon: physical recycling to polymers** | | | | |
| Fulgar with the MSC process | Post-industrial waste | Q-Nova®, 50/50 regenerated/virgin nylon 6,6 fibre | Unknown | [51] |
| **Nylon: chemical recycling to monomers** | | | | |
| Aquafil with Econyl® technology (depolymerisation to caprolactam) | Nylon 6 fishing nets, carpets, post-industrial textiles | Econyl® yarn, nylon 6 | Commercial, Italy | [52] |

#### 4.1.4. Polyurethane

Recycling elastane, being a polyurethane and polycondensation polymer, is a challenge. At present, no methods are available on a pilot or demo scale. Elastane is usually present in (women's) fashion in only a very small fraction, making recycling less interesting. Elastane is usually not recycled but can be removed from the fabric to facilitate the recycling of other fibres, e.g., by solvolysis methods suitable for polycondensation polymers [35,53].

#### 4.1.5. Polyolefin

Polyolefins such as polyethylene (PE) or polypropylene (PP) are thermoplastics and can, in theory, be physically recycled by melting and re-spinning to new fibres and yarns, but for apparel textiles, we did not find any examples. Polyolefins are addition polymers and can therefore not be recycled by chemical means, e.g., depolymerised to their monomers by solvolysis methods, as the chemical bonds in the polymer cannot be broken by these methods. These polymers can be degraded through a free radical mechanism at high temperatures, but this does not result in the formation of re-usable monomers. Instead, heterogeneous mixtures of gasses, liquids and tar are produced [54] that may be used as input for the chemical cracking process and thus add to the production of renewable bulk chemicals.

#### 4.1.6. Polyacrylic

Polyacrylics can be mechanically recycled, comparable to wool. The process involves colour sorting, cleaning, unravelling and spinning again [49]. Polyacrylics cannot be melted, and although they can be dissolved, presumably no physical recycling methods are under development. Polyacrylics are formed by addition polymerisation and can therefore not be depolymerised by solvolysis methods.

#### 4.2. Feasibility Assessment for Main Textile Polymers and Fibres

Recycling options for each type of textile fibre is a complex topic, as each fibre has its own optimal recycling strategy. A feasibility assessment of various recycling routes for the six polymers and the main textile fibres discussed is shown in Table 6. The emphasis here is on options for recycling textile fibres back into textile fibres.

**Table 6.** Most achievable (green), less achievable (orange) and not achievable (red) recycling options for main polymer groups.

| Classification of Textile Recycling | Cellulose | | Polyamide | | Polyester | Polyurethane | Polyolefin | Polyacrylic |
|---|---|---|---|---|---|---|---|---|
| | Natural | | Condensation | | | | Addition | |
| | Cotton, Linen | Viscose, Lyocell | Wool | Nylon | PET | Elastane | PP, PE | Acrylics |
| Fibre recycling | green | orange | green | orange | orange | red | orange | green |
| Polymer recycling | green * | green | red | green | green | red | orange | orange |
| Monomer recycling | orange | orange | red | green | green | orange | red | red |

\* To another type of fibre.

For natural polymers, we see multiple options for fibre and polymer recycling. The fibre recycling of natural staple fibres such as cotton is possible by mechanical means, provided that the resulting fibre length is sufficient to make yarn spinning possible. The polymer recycling of cellulose by physical means is possible by processes where the cellulose is dissolved and spun into new textile fibres. For wool, the recycling of fibres by mechanical means is the only option.

Condensation polymers such as PET and nylon can be recycled by physical and chemical methods. Polymers can be recycled by melting the material at high temperatures and re-spinning it into filaments. When high-quality products are desired, only highly purified materials can be used. Another viable option is recycling the monomers by chemical means. The input for these processes can be less pure material while the end-

product is of virgin quality, but the monomers need to be 100% pure before repolymerisation can take place. Elastane has limited options for recycling.

In the category of addition polymers, the options are limited. For polyacrylics, recycling the fibres by mechanical means might be an option. The fibres can be spun into yarns to produce knitwear.

### 4.3. The Challenge of Blended Textiles

Contemporary textiles are increasingly composed of a variety of fibres, as single-fibre textiles often do not fulfil the requirements of today's fashion. Fibre blending, combining two or more fibre types to form a new yarn or fabric, can combine the best qualities of each fibre. In this way, the functionality of the fabric is improved [10]. A well-known example is the presence of a few per cent of elastane in jeans to improve wearing comfort [55]. Another example is a blend of PET and cotton to combine strength and comfort and to reduce the price, as polyester fibres are cheaper than cotton fibres [56]. Another reason can be to improve ease of processing, as synthetic fibres can be produced in a broader range of fibre lengths than the relatively short length of natural staple fibres, making spinning easier [56].

A wide range of fibre combinations is used nowadays. However, reliable data are hard to obtain and are mostly limited to analyses of post-consumer textiles, for example, through infrared technology. For sustainability and end-of-life options, fibre blending might not be such a good idea. In this section, we explore recycling options for blended fabrics and garments.

#### 4.3.1. Post-Industrial and Post-Consumer Waste Streams

The most significant issues with blended textiles relate to post-consumer waste streams in the form of garments. These garments often consist of multi-material fibre compositions, making recycling very complicated. These different types of fibres need to be separated for recycling, which is difficult or even impossible [8]. Post-industrial waste streams are often well-defined and more homogeneous, as they can be collected as fibres, yarns or fabrics, thus well before the garment is produced. Post-industrial sources, therefore, have a much greater chance of achieving cost-effective, high-quality recycling than post-consumer streams [57], and some of these recycling methods are already implemented. However, according to the Ellen MacArthur Foundation, less than 1% of the material used to produce clothing is recycled into new clothing. This includes recycling after use, as well as recycling of factory offcuts [2].

All post-consumer textiles are collected and sorted in the ideal situation. However, in reality, this amount is never reached. For the Dutch situation, around 45% (136 kton) is collected separately, whereas 169 kton ends up as residual waste [58]. The amount of discarded textile per capita increased from 4.2 to 4.7 kg/y over the past 10 years [59]. Approximately half of the collected textiles are suitable for the second-hand market (mostly abroad), and the remainder is only suitable for recycling. Due to the rise of fast fashion and the decline of the second-hand market, the amount of textile suitable only for recycling (non-wearable fraction) increases every year [59].

The composition of this non-wearable fraction is not well-defined. Sorting companies such as *Leger des Heils* (Salvation Army) and Sympany sort by hand into main categories such as denim, white cotton, coloured cotton and sweaters [60]. Advanced sorting technologies such as the Fibresort [61] can sort textiles based on composition (e.g., wool, cotton, nylon, PET) and even colour but can only process mono-materials. Although we cannot support this with data, we assume that single-fibre garments are outnumbered by blended textiles, creating a big problem for textile recycling. Converting post-consumer mono-material textile fibres into new textiles is possible on a limited scale. As can be concluded from Tables 3–5, the options are even more limited for blended textiles. Table 7 shows initiatives by companies and start-ups that work on the challenge of recycling blended textiles. Most of these initiatives deal with an input stream of cotton and PET and produce a stream of PET monomers and regenerated cellulose fibres.

**Table 7.** Initiatives of recycling of blended textiles (partly adapted from [19].

| Company | Input Stream | Product | Status | Ref. |
|---|---|---|---|---|
| HKRITA/H&M Foundation with hydrothermal method | Cotton/PET | New textile fibres | Pre-industrial size facility in Hong-Kong | [62] |
| Worn Again with dissolution of PET and cellulose | Cotton/PET | Cellulose pulp for regenerated cellulose fibre and PET | Industrial demonstration plant to be launched in 2021 | [63] |
| Blend Re:Wind (Rise and Chalmers University) | Cotton/PET | Viscose filaments and PET monomers | Sweden, status unknown | [64] |
| Sodra with Once More | Cotton/PET | Cellulose pulp for textile | Sweden, commercial | [65] |
| Tyton Biosciences | Cotton, PET, polycotton | Cellulose and PET monomers | USA, status unknown | [66] |
| Block Texx | Cotton/PET | Cellulose pulp and PET | Australia, status unknown | [67] |
| Resyntex with monomer recycling | Cotton, nylon, PET, wool | Monomers (glucose, TPA, EG) protein hydrolysates, polyamide oligomers | Closed EU-project | [68] |
| Trash2Cash with polymer recycling | Blended textile and paper waste | Regenerated cellulose | Closed EU-project | [69] |

### 4.3.2. Common Textile Blends

Although the composition of post-consumer textiles is not well-defined, based on our experience, some combinations are frequently used. In this paragraph, the main fibre blends are discussed for each group of polymers, including possible recycling strategies.

A common blend is a cotton with a few per cent of elastane. Mechanical recycling of cotton would be the most viable option, but elastane may cause problems during mechanical processing. The cotton fibre may also be of inferior quality after use, making the conversion of cotton into regenerated cellulose fibre a better option. The best-case scenario is when the elastane fraction ends up in a solid residue while cellulose is dissolved and spun into a new fibre. The same holds for blends of elastane with other cellulose fibres such as linen, viscose or lyocell. Whether this is feasible depends on the process applied, and further investigations are needed.

A blend of cotton and PET is often used for workwear, and production volumes are high. Several approaches can be followed to deal with polycotton waste streams [70]. In general, approaches where the cellulose remains intact as much as possible are preferred, as only PET monomers can be polymerised into a new polymer. The challenge is to use mild conditions to preserve the cellulose fraction, as most of the solvolysis methods of PET will certainly degrade cellulose into smaller polymer fractions.

PET is blended with almost all other textile fibres mentioned in this publication. Although PET recycling from bottles is well underway, recycling blended textile fibres is less developed. Polymer recycling by melting and spinning into new textile fibres is less achievable due to contaminations, leaving monomer recycling as the only option. The results from a study by GreenBlue indicate that, for the chemical recycling technologies being evaluated, a minimum purity level of 70–80% of PET is required for an economically feasible process [57].

Natural polyamide fibres (i.e., wool and silk) are often used as a mono-material, and blends are limited. Wool can be combined with acrylic and still look like a woolen garment. Depending on the composition, it might be possible to also mechanically recycle blends of

wool and acrylic, as is done with purely woolen garments. The manufactured polyamides, nylons, are frequently combined with other fibres such as cotton (e.g., in lace) or viscose (e.g., in knitted fabric). Similar approaches as for the combination of cotton and PET could be taken here.

## 5. Conclusions

Over the coming decades, a shift of the global textile value chain from a linear to a circular model is foreseen, driven by new regulations, the desire for resource efficiency, cost concerns and consumer demands. The textile industry is not prepared for this transition, as it lacks circular design competencies and efficient ways to recycle textile residues. For the textile industry to become more sustainable, knowledge of the origin and production of resources is important. Recycled feedstock, in addition to virgin renewable resources, is expected to form a significant part of the future resources to be used.

Textile fibre classifications are often made by origin and not by polymer type. Fibre properties are always described from the initial application and their behaviour during their lifetime through exposure, use or maintenance. Recyclability has not been a relevant variable until now.

For recycling purposes, we classified textile fibres based on their main chemical bonds, as fibres with the same kinds of bonds usually have similar chemical and physical characteristics. We distinguished cellulose, polyamide, polyester, polyurethane, polyolefin and polyacrylic as the principal polymer groups for textile fibres. In addition, a new classification of textile recycling technologies was designed based on the level of disassembly (fibre, polymer, monomer) combined with mechanical, physical and chemical recycling.

We showed that for fibre recycling by mechanical means, fibre length is the most crucial parameter. This type of recycling works best for cotton, linen, wool and acrylics. The quality of the recycled product is often lower than that of virgin resources and is highly dependent on the quality of the input stream. For polymer recycling by physical means, the molecular weight of the polymer and the ability to dissolve or melt are important. This type of recycling is best applied to cotton, linen, viscose, lyocell (dissolve) and nylon and PET (melt). The quality of the recycled product can approach virgin quality. For monomer recycling by chemical means, the ability to depolymerise the polymer to its monomeric building blocks is key, combined with an efficient recovery. This type of recycling is only suitable for polycondensation polymers such as nylon and PET. The most important advantage is that virgin quality can be achieved. Disadvantages include high costs and the needed large-scale production for economic feasibility.

For consumers, the choice between virgin and recycled products depends on the perceived benefits or disadvantages. Recycled content is often a synonym for low quality. However, here we showed that the final quality depends strongly on the type of fibre and technology used and that each fibre has its prefered recycling technology.

Textile recycling (especially post-consumer) is still in its infancy and will be a major challenge in the coming years. In general, there are good recycling options for mono-material residue streams, but the real challenge lies in blended textiles. The volume and composition of blended textiles allocated for recycling are often unknown, which was a limitation for this study. The plethora of fibre combinations added to this problem. Recycling blended textiles is possible to a limited extent with the currently available recycling techniques. However, when recycling is technically complicated, energy-consuming and expensive, it will most likely not become a profitable business, especially in combination with cheap virgin materials. For blended textiles, the perspective is promising for fibre blends within a single polymer group, while combinations of different polymers are undesirable.

To be able to Recycle, we must Rethink. Outdoor apparel brands have been experimenting with different design and material selection strategies that enhance the recyclability of their products. Examples are fabrics based on one type of fibre, fibres with better recyclability profiles, and creating a market demand for recycled materials [57]. Adoption of these

approaches by other players in the textile industry is urgently needed, but initiatives are still scattered and small scale.

The intricate blending of different types of fibres that require different recycling strategies should be prevented, and the use of fibres that originate from renewable resources combined with good recycling options should be encouraged. Our methodology can help stakeholders in the textile industry to critically assess their production methods and the materials they apply. If the approaches we propose become more widespread, the recycling of fashion can grow, and the sustainability of the textile and fashion industry will improve.

**Author Contributions:** Conceptualisation, P.H. and H.B.; methodology, P.H. and H.B.; writing—original draft preparation, P.H.; writing—review and editing, H.B. and M.S. All authors have read and agreed to the published version of the manuscript.

**Funding:** This research was funded by Wageningen University and Research Knowledge Base Program *Towards a circular and climate neutral society* (KB34), project Recycling and end-of-life strategies for sustainability and climate (KB-34-011-001).

**Conflicts of Interest:** The authors declare no conflict of interest.

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
