# Peer review of "Textiles for Circular Fashion: The Logic behind Recycling Options"

_sustainability, doi:10.3390/su13179714_

Round 1

Reviewer 1 Report

My main recommendation is in regards of the research design, research questions hypothesis and methods that must be clearly developed. Authors should present their research methodology as a whole. We must have research objectives and/or research questions more clearly stated at the beginning of the section 2.

Line 89: Could you please detail more in details the feasibility assessment. How was it constructed? What is the development methodology. 

Line 187 The figure is clear and provides a clear understanding of the processes through linear and circular prisms.

Overall, the article is much relevant, well structured and brings a great overview of classification of fibre by types and recycling options and initiatives.

The concluding perspectives could be fleshed out a little bit more. 

Reviewer 2 Report

The manuscript addresses a relevant topic, that is, the textiles for circular fashion and recycling options. Although it has the potential to be considered for publication due to its intrinsic quality and relevance, there is a substantial flaw that needs to be overcome, that is, the lack of theoretical background. Thus, it is suggested to create a new section 2. Theoretical background, where the relevant previous studies need to be reviewed, including the one already published in your target journal. The original section devoted to Methods needs to be revised, by merging all the sub-sections.

In the concluding remarks, it is also suggested to include the limitations of the analysis, as well as policy and managerial implications, especially, focused on circular economy and sustainable business models applied to the textile industry.

Reviewer 3 Report

Dear authors, the recycling of textiles is a hot topic in the extant literature and papers focusing on it are welcome. However, your work must be improved before proceeding with its publication:

1) Abstract lacks in several elements (e.g. main gaps assessed/considered by your work, research method, main results, future trends)

2) Section 1 must better clarify what are the main issues coming from the literature and which of them will be afforded by your work. In addition, the structure of the paper shoud be presented at the end of the section.

3) Pictures and tables must be centred within the text

4) Section 2 must be improved in details about your research method, by giving more links with similar works in the same sector.

5) Section 3 must be splitted in 2 sections, one dedicated to results and one dedicated to discussions on recycling options. In addition, the content must be shortened a lot.

6) Conclusions must be better linked with results and discussions, by offering more relevant information.

7) In order to enforce your references in terms of circular business models there are lots of papers in literature. For example, please cite: Rosa et al 2019 "Circular Business Models versus circular benefits: An assessment in the waste from Electrical and Electronic Equipments sector"; Rosa et al 2019 "Towards Circular Business Models: A systematic literature review on classification frameworks and archetypes"

Round 2

Reviewer 2 Report

The revised version of the manuscript was substantially improved. Nevertheless, before publication, it is recommended to implement the previous suggestion for revision purposes of section 2. Methods.

Author Response

For section 2 all sub-sections were merged. In addition, novelty of the work and hypoyhesis are added in section 1 (see lines 94-97).

Reviewer 3 Report

Dear authors, thanks for having followed my suggestions. Before proceeding with publication, I kidly ask you to shorten the discussion and conclusion sections.

Author Response

section 1 was completed with novelty of the work and our hypothesis. section 2 was improved by merging all the subsections (as requested by other reviewer). discussion section 4 was improved by eliminating text that was more background information and not adding to the discussion. #words was thereby reduced from 9356 to 8563 words.